# Isolation and Characterization of Cryotolerant Yeasts from Fiano di Avellino Grapes Fermented at Low Temperatures

**DOI:** 10.3390/foods12030526

**Published:** 2023-01-24

**Authors:** Ernesto Petruzziello, Giuseppe Blaiotta, Elisabetta Pittari, Paola Piombino, Maria Aponte

**Affiliations:** 1Division of Vine and Wine Sciences, Department of Agricultural Sciences, University of Naples Federico II, Viale Italia, 83100 Avellino, Italy; 2Task Force on Microbiome Studies, University of Naples Federico II, Via Cinthia, 80126 Naples, Italy; 3Division of Microbiology, Department of Agricultural Sciences, University of Naples Federico II, Via Università 100, 80055 Naples, Italy

**Keywords:** grapes, wine, fermentation, yeast, diversity, cryotolerant, *Saccharomyces*, *S. kudriavzevii*, *S. paradoxus*, VOCs

## Abstract

A fermentation of Fiano di Avellino grape must was carried out at 9°C with the aim of selecting cryotolerant yeast strains and testing their fermentative performances and volatile production following molecular characterization. A total of 20 yeast cultures were isolated at different fermentation stages. Based on molecular identification and characterization, *Metschnikowia* (*M*.) *pulcherrima*, *Hanseniaspora* (*H*.) *uvarum*, *Staremerella* (*St*.) *bacillaris*, *Saccharomyces* (*S*.) *cerevisiae*, *S. kudriavzevii*, and *S. paradoxus* were found to be the yeast species dominating the fermentation. *S. paradoxus* has been rarely isolated in vineyards and never in the cellar environment. Moreover, in this study, *S. kudriavzevii* is detected for the first time in vine-wine environments. Both *S. kudriavzevii* and *S. paradoxus* co-occurred with *S. cerevisiae* when grapes were micro-fermented at low temperatures. The growth kinetics of the three species were greatly affected by the fermentation temperature. As a consequence, Fiano wines obtained with *S. kudriavzevii* and *S. paradoxus* significantly differed from those made by *S. cerevisiae* in terms of chemical and volatile composition.

## 1. Introduction

Low-temperature wine fermentation enhances the freshness and fruity notes of the final product by preserving both the varietal and fermentative aromas to a greater extent [1,2]. Despite the quality improvement, low fermentation temperature has some negative implications: (1) longer fermentation time; (2) higher risks of oxidation; (3) major management and energetic costs; (4) increased persistence of non-*Saccharomyces* yeast. At low temperatures, the rate of ethanol production, as well as the growth of *Saccharomyces* (*S.*) *cerevisiae,* is lower, whereas the ethanol resistance of non-*Saccharomyces* is higher. As a consequence, the probability of stuck and sluggish fermentations increases. Therefore, the wine industry is clearly interested in using yeast strains with an enhanced capability to ferment at low temperatures. In the *Saccharomyces sensu stricto* complex, some cryotolerant strains of *S. kudriavzevii*, *S. uvarum,* or *S. eubayanus* were so far isolated and characterized. However, due to the greater ethanol sensitivity than *S. cerevisiae*, their application in the cold winemaking process is limited [3]. By contrast, in wine environments, cryotolerant natural hybrids (*S. uvarum* × *S. cerevisiae*, *S. cerevisiae*, and *S. kudriavzevii*) have been described [3,4]. As summarized by García-Ríos [5], these hybrids obtain their physiological capability from both parents. Hybrids might have inherited the ability to grow at high temperatures (30–37 °C) and the ethanol tolerance from *S. cerevisiae* as well as the ability to grow at low temperatures (10–16 °C) from the other parent (*S. kudriavzevii*, *S. uvarum*, and *S. eubayanus*). Artificial interspecific hybrids into the *Saccharomyces* genus were also proposed as a possible biotechnological solution for improving cryotolerance in wine yeasts [5]. 

Given these facts, the industry is driven to select yeast strains with improved low-temperature fermenting abilities. Naturally, cold-tolerant strains of the *Saccharomyces* genus, such as *S. kudriavzevii*, *S. uvarum*, or *S. eubayanus*, may be employed for low-temperature fermentations. However, they might not be as efficient for alcoholic fermentation because they typically have higher ethanol sensitivity than *S. cerevisiae*. In this study, natural cryotolerant yeasts were isolated and characterized from Fiano white grapes, one of the most significant vine cultivars in the Irpinia wine district (Avellino Province, Campania Region, Italy), used to make the well-known Fiano di Avellino DOCG (Appellation of Controlled and Guaranteed Origin) wine. Grapes were harvested and crushed in sterile conditions and allowed to ferment in an old cellar at a temperature that was naturally set at 9–10 °C in order to focus the search on yeast strains that are well acclimated to low temperatures. The isolated yeast cultures were identified and characterized by DNA-based techniques and evaluated for features of technological interest. Selected cultures were employed for microfermentation trials. 

## 2. Materials and Methods

### 2.1. Origin of Grape Samples and Fermentation Conditions

Fiano grapes were collected from a vineyard (N 40° 58′ 16,59716″, E 14° 44′ 28,34808″) located in Sant’Angelo a Scala (Irpinia district) in the first decade of November 2021. Grapes (about 5 kg) were harvested using sterile gloves and plastic bags (previously weighed) and immediately transferred to the laboratory. Grapes were aseptically crushed in the collecting bags, weighed, and transferred in a sterile 10 L steel container. After the addition of potassium metabisulfite (100 mg/kg), ammonium phosphate (200 mg/kg), and thiamine (0.6 mg/kg) [6,7], the must batch was left to ferment in a historic cellar at 9–10 °C. 

### 2.2. Fermentation Monitoring and Yeast Isolation

At the beginning (T0 day) and during fermentation (T7, T14, T21, and T28 days), the main oenochemical parameters (°Babo, pH, total acidity, and alcoholic degree), as well as yeast populations, were monitored. Sugar content was determined by using Babo mustimeter (Klosterneuburg). pH meter Basic 20 (Crison) for pH and total acidity determinations. Total acidity was expressed as g/L of tartaric acid (mL of 0.25 N NaOH to neutralize 25 mL of must/wine × 0.75). The alcoholic degree was evaluated by using the Malligand Ebulliometer. All analyses were performed in triplicate.

For viable yeast counts, samples were serially diluted in sterile saline solution (NaCl 8.5 g/L, peptone 1.0 g/L, tween 80 0.5 g/L) and spread-plated on WL-nutrient agar (dextrose 50 g/L, yeast extract 4 g/L, casein enzymic hydrolysate 5 g/L, monopotassium phosphate 0.55 g/L, potassium chloride, 0.425 g/L, calcium chloride, 0.125 g/L, magnesium sulfate 0.125 g/L, ferric chloride 0.0025 g/L, manganese sulfate 0.0025 g/L, bromocresol green 0.022 g/L, agar, 20 g/L, pH 5.5 ± 0.2, Oxoid, Basingstoke, UK) supplemented with 100 mg/L of chloramphenicol (Fluka, Milan, Italy). After incubation at 28 °C for 5 days, countable plates (15–150 colonies/plate) were used for viable counts and yeasts’ isolation. Colonies showing different morphology and/or color were all selected independently by their number. Cultures were purified by repetitive streaking on WL-nutrient agar. Yeast cultures were preserved on WL-nutrient agar slants and freeze-dried on malt extract broth (Oxoid) containing 20% (*w/v*) of glycerol (Fluka). Before each test, strains were cultured twice in YPD (yeast extract 10 g/L, peptone 20 g/L, dextrose 20 g/L, Oxoid).

### 2.3. Yeast Molecular Identification and Typing

DNA was isolated by using the method reported by Aponte and Blaiotta [6]. Preliminary identification was achieved by ITS (Internal Transcribed Spacer) rDNA region analysis coupled to ITS-RFLP (ITS-Restriction Fragment Length Polymorphism) by using *Hae*III as a restriction enzyme. The taxonomic affiliation of representative strains of each ITS-RFLP group was determined by ITS region sequencing. All isolates belonging to the genus *Saccharomyces* spp. were typed by Interdelta [8] and DAN4 [9] analyses. Moreover, selected *Saccharomyces* spp. strains were characterized by RFLP analysis or/and sequencing of nuclear genes CAT8, CYR1, GSY1, MET6, and OPY1, as described by Gonzalez et al. [4]. Specifically, for RFLP analyses, the following restriction endonucleases were used: *Cfo*I for CAT8 gene; *Hae*III for CYR1 and OPY1 genes; *Msp*I for GSY1 gene; *Hinf*I for MET6 gene. DNA from commercial strain EC1118 (LALVIN) was also analyzed in this phase. 

### 2.4. Technological Characterization of Saccharomyces spp. Strains

Ethanol tolerance was evaluated in YPD broth at pH 3.20 (adjusted with tartaric and malic acids 1:1) containing 100 mg/L of potassium metabisulphite and increasing ethanol concentrations ranging from 4 to 14% (*v*/*v*). Growth was evaluated after incubation at 28 °C for 72 h by spectrophotometry at white light (600 nm). Production of hydrogen sulfide (H_2_S) was estimated on Biggy agar (Oxoid) as previously described [10]. Beta-glucosidase activities were evaluated on media containing 4-methylumbelliferyl-b-D-glucopyranoside (MUG), arbutin (ARB), esculin (ESC), or cellobiose (CEL) (Fluka), according to the method proposed by Fia et al. [11] and Hernandez et al. [12]. Lipolytic activity on tween 80 and proteolytic activity on milk proteins was assessed according to Slifkin [13] and Fadda et al. [14], respectively.

Growth kinetics at different temperatures (10, 14, 18, and 28 °C) were evaluated in YPD broth at pH 3.2. At 10 °C, the commercial strain EC1118 (LALVIN) was used as control. The growth was monitored by spectrophotometric determinations at 600 nm (OD600).

### 2.5. Microfermentation Trials

The standardized procedure proposed by Romano et al. [15] was followed for microfermentation experiments. Strains cultured twice in YPD medium were used to inoculate (about 6 Log CFU/mL) Fiano must (22 °Brix, total acidity 6.36–6.42 pH 3.17–3.20) sterilized by tyndallization (100°C for 3 min, 3 times) in 250 mL Erlenmeyer flasks closed with a Müller valve filled with sulphuric acid. The must was supplemented with potassium metabisulfite (100 mg/kg) and ammonium phosphate (200 mg/kg) before inoculum. During incubation, at 9–10 °C, flasks were handle stirred for 30 s every 12 h. Weight loss due to CO_2_ escaping from the system was quantified for the fermentation kinetics monitoring. Fermentation was considered concluded when no weight loss was recorded within 24 h. Fermentation vigor (FV) was expressed as grams of CO_2_ produced in 100 mL of Fiano must during the first 5 days of fermentation, while fermentation power (FP) was expressed as the alcoholic degree (% *v*/*v* = g CO_2_ produced/100 mL × 1.25) reached at the end of fermentation. Each trial was performed in triplicate. At the end of fermentation, total sugars (glucose plus fructose), acetic acid, total acidity (expressed in grams/L of tartaric acid), and total and free SO_2_ were quantified by using a Dionysos 100 multiparametric analyzer (Sinatech, Fermo, Italy). For each parameter, a specific kit (Sinatech, Fermo, Italy) was used. Moreover, concentrations of glucose, fructose, glycerol, ethanol, as well as tartaric, malic, and succinic acids, were evaluated by HPLC (High-Performance Liquid Chromatography) as previously described [10].

### 2.6. VOCs Analysis

For VOCs extraction, 100 mL of wine was extracted by applying a liquid-liquid extraction method described by De Filippis et al. [16]. Each extraction was carried out in triplicate. For High-Resolution Gas-Chromatography/Mass Spectrometry (HRGC/MS) analysis, 2 µL of the organic extract was injected in splitless mode, while the injection port of a GC/MS-QP2010 quadrupole mass spectrometer (Shimadzu, Shimadzu Corp., Kyoto, Japan) was maintained at 250 °C. The GC/MS was equipped with a DB-WAX column (60 m, 0.25 mm i.d., 0.25 μm film thickness; J&W Scientific Inc., Folsom, CA 95360, USA). The carrier gas was helium (1.3 mL/min), and the temperature program used was 0 °C for 5 min, raised to 220 °C at a rate of 2 °C/min, and held for 20 min at the maximum temperature. Electron impact mass spectra were recorded with ion source energy of 70 eV, while the temperature was kept at 230 °C. The peak areas were measured using a GC/MS solution program Shimadzu version 2.30 (Shimadzu Corp., Kyoto, Japan). Compounds concentrations (semiquantitative) and identification were computed and performed as previously reported [17]. Concentrations of isolated compounds were expressed as a ratio of the response of each compound against the response of the internal standard.

In a few cases, the pure chemical standard was not available, and the identified compounds were labeled as tentative (t). 

### 2.7. Statistical Analysis

Significant differences among data were tested by ANOVA (Tukey). Principal component analysis (PCA) was employed to test the relationship (Pearson correlations) between volatiles and wine samples. The significance level was *p* ≤ 0.05 throughout the analyses. Data elaboration was carried out using XLStat (Addinsoft Corp., Paris, France, version 2012.6.02), an add-in software package for Microsoft Excel (Excel 2013).

## 3. Results

### 3.1. Fermentation Monitoring and Yeast Isolation

The main oenochemical parameters detected during the fermentation of Fiano grape must at 9–10 °C are reported in Table 1. Fermentation started with a natural yeast microflora of about 10^6^ CFU/mL. Yeast loads reached the maximum (about 10^8^ CFU/mL) after seven days and then slowly declined. After 28 days, sugars were completely metabolized (0 Babo). The wine showed an alcoholic degree of 13.6% *v/v*, a little lower than expected, a pH of 3.88, and a total acidity of 6.04 g/L. In order to characterize the dominant yeast microflora, colonies were collected at each fermentation phase from countable agar plates: two from must (F0), five at 7 days of fermentation (F7), four at 14 days (F14), four at 21 days (F21) and five at 28 days (F28).

### 3.2. Yeast Molecular Identification and Typing of Isolates

All isolates were identified and characterized by means of molecular tools. As Table 2 shows, only four different ITS profiles were identified: 380 bp (showed by two isolates), 480 bp (one isolate), 750 bp (two isolates), and 850 bp (two isolates). Based on ITS-RFLP analyses and ITS sequencing, the two isolates with ITS of 380 bp were identified as *Metschnikowia* (*M*.). *pulcherrima*, the two isolates with ITS of 750 bp as *Hanseniaspora* (*H*.) *uvarum*, and the isolate with ITS of 480 bp as *Staremerella* (*St.*) *bacillaris* (Table 2 and Table 3). Isolates with ITS of 850 bp (*Saccharomyces* spp.) showed the same ITS-RFLP patterns with both *Cfo*I and *Hinf*I restriction enzymes (Table 2). By contrast, with *Hae*III, two isolates (F4-72 and F14-72) showed a profile with only three bands (500, 230, and 150 bp, respectively) (Table 2). All Saccharomyces spp. isolates were analyzed by means of molecular markers Interdelta and DAN4. By Interdelta typing, only three different profiles were retrieved: pattern reported as A characterized 11 isolates, and pattern B and pattern C were exhibited by only two isolates: F7-72 and F14-62, and F21-62 and F28-52, respectively (Table 2). These four isolates did not produce any pattern when analyzed by DAN4 molecular marker (Table 2). Therefore, based on ITS-RFLP, Intedelta, and DAN4, *Saccharomyces* spp. strains could be gathered in three different biotypes: (*i*) F7-72 and F14-72; (*ii*) F21-62 and F28-52; (*iii*) all the remaining. One isolate of each group (F4-72, F21-62, and F14-62, respectively) was further analyzed by other molecular markers. By means of ITS sequencing, the three strains were identified as *S. kudriavzevii*, *S. paradoxus*, and *S. cerevisiae*, respectively (Table 2 and Table 3). Moreover, to exclude/confirm their hybrid nature, RFLP analysis and sequencing of some nuclear genes were performed. Data were compared by those obtained by *S. cerevisiae* EC1118 or available in open-source databases. 

*S. cerevisiae* F14-62 showed an RFLP profile identical to that of EC1118 (Table 2). The other two strains (F7-72 and F21-62) showed RFLP profiles similar to those reported for *S. kudriavzevii* and *S. paradoxus* by Gonzalez et al. [4], respectively. All patterns were straightforward, and no mixed profile could be seen, allowing interspecific hybrid strains to be ruled out. To clear up any confusion, all the genes of the strains *S. kudriavzevii* F7-72 and *S. paradoxus* F21-62 were sequenced. Sequencing results of nuclear genes CAT8, CYR1, MET6, GSY1, and OPY1 supported the identification of strains F7-72 and F21-62 as *S. kudriavzevii* and *S. paradoxus*, respectively (Table 3). 

### 3.3. Technological Characterization of Saccharomyces spp. Yeast Strains

The *Saccharomyces* spp. strains *S. kudriavzevii* F7-72, *S. cerevisiae* F14-62, and *S. paradoxus* F21-62 were selected as the population’s representatives. In YPD broth with a pH of 3.20, 100 mg/L of potassium meta-bisulfite, and 13% (*v/v*) ethanol, all strains could thrive. Moreover, on Biggy agar, all of them proved to be low H_2_S producers (Data not shown). On media containing 4-methylumbelliferyl-b-Dglucopyranoside, arbutin, esculin, or cellobiose, no strain displayed beta-glucosidase activity, and none of them expressed lipolytic or proteolytic activity (Data not shown). On the other hand, by comparing growth kinetics at various temperatures, particularly when higher than 14 °C, variations across strains were revealed. All strains displayed similar trends at 10 °C, but as the fermentation temperature was raised, *S. cerevisiae*’s development was noticeably higher (Figure 1). 

### 3.4. Microfermentation Trials

Microfermentations were carried out in tyndallized Fiano must at 9–10 °C. The three non-*cerevisiae* species had somewhat similar CO_2_ development, despite the fact that *S. paradoxus* (F21-62) produced significantly less CO_2_ at each testing point (Figure 2). Actually, compared to *S. kudriavzevii* (F7-72) and *S. cerevisiae* (F14-62), *S. paradoxus* (F21-62) exhibited a lower FV (5 days) and FP (end of fermentation, 35 days) (Figure 2).

Low significance differences were also observed when the main oenochemical characteristics of the produced wines were evaluated (Table 4). *S. paradoxus* (F21-62) produced less CO_2_ during fermentation (Figure 2), and as a matter of fact, the residual sugar content in the wine was the highest (4.5 g/L). However, even though it is less than that of *S. cerevisiae* F14-62 8 (about 14%), the ethanol content was higher than that of the wine made with *S. kudriavzevii* F7-72 (about 11%), which had a residual sugar content of about 2 g/L. (Table 4). A distinctive characteristic of the wines produced by *S. paradoxus* (F21-62) and *S. kudriavzevii* (F7-72) was the higher glycerol content.

### 3.5. VOCs Analysis

Forty-three VOCs were identified by HRGC/MS: 15 esters, 11 alcohols, nine acids, and eight miscellaneous compounds (Table 5). According to the ANOVA (Tukey, *p* < 0.05), *S. kudriavzevii* wine samples were characterized by an overall higher VOC production. Wines produced with *S. cerevisiae* were the richest in 3- + 2-methyl-1-butanol and isobutyl alcohol. They had a significantly higher level of isoamyl acetate, an ester that is the most powerful wine fruity aroma with a banana note. 

Moreover, *S. cerevisiae* wines were the samples showing the lowest levels of some volatile acids. Differently, *S. kudriavzevii* samples showed significantly higher amounts of some volatile acids (isobutyric, butyric, nonanoic, and decanoic acids) and important contributors to the fruity aroma of wine, such as ethyl butyrate, hexanoate, and decanoate. *S. kudriavzevii* wines were also the richest in beta-phenylethyl acetate and beta-phenyl ethanol. *S. paradoxus* showed an intermediate behavior between *S. kudriavzevii* and *S. cerevisiae*. Indeed, for very few VOCs, significant differences were found between the three wines. Amongst these significant differences, two compounds (i.e., 1-butanol and 3-methyl-1-pentanol) showed the highest concentrations in *S. paradoxus* samples, and three miscellaneous VOCs (2-pentanone, beta-citronellol, and benzothiazole) showed the lowest concentrations in *S. paradoxus* wines. These wines contain similar levels of the smoky 4-vinyl guaiacol to that of *S. kudriavzevii* and significantly higher compared to *S. cerevisiae*, which produced more methionol, while *S. kudriavzevii* more acetoin (Table 5). 

A principal component analysis was carried out considering the fermentative replications for each wine (observations) and the volatile compounds (variables) (Figure 3). The first two components, accounting for 72.15% of the variance (50.45% and 21.70% on F1 and F2, respectively), clearly distinguish the different wines in three different regions of the biplot. The biplot mainly opposes samples produced by the fermentation with *S. cerevisiae* F14-62 to those fermented by *S. kudriavzevii* F7-72 and *S. paradoxus* F21-62 (Figure 3a). All the fermentative replicates are grouped on the PCA according to the fermentative yeast showing good repeatability of the experimental replicates, except for the sample F7-72_1, then excluded in the following analyses. On the positive semiaxis of the first component, most of the identified VOCs are well correlated to each other and F7-72 wines fermented with *S. kudriavzevii*. Among these VOCs, there are most of the fermentative volatiles, even if amyl alcohols (3- and 2-methyl-1-butanols) and the relative isoamyl acetate are well correlated with *S. cerevisiae* (F14-62) samples along the opposite semiaxis (Figure 3b).

## 4. Discussion

Grape musts are characterized by a complex microbial ecology, including filamentous fungi, yeasts, and bacteria. Some microbial species are found only in musts before the onset of fermentation, while other species, such as yeast, lactic, and acetic acid bacteria, may survive and/or grow during the winemaking process and, due to their different physiological characteristics, mightily affect the wine quality. In must, these microorganisms are subjected to a selective pressure exerted by several factors, including high sugar content, high acidity, nutrient availability, low oxygen tension, increasing ethanol concentrations, and the presence of specific inhibitors such as SO_2_, botriticin, and medium chain fatty acids [18]. Moreover, among abiotic factors, fermentation temperature plays a crucial role in the growth and fermentation performances of different yeast species [19]. Fermentation at low temperatures is becoming a trend for the enhancement of wine aroma ‘freshness’ [1,2,20]. In this study, the yeast population dynamics during cryo-fermentation (9 °C) of a Fiano grape must were evaluated. Following a starting phase that was dominated by *M. pulcherrima* and *H. uvarum*, a mixed population of *Saccharomyces* spp., comprising *S. kudriavzevii*, *S. paradoxus*, and *S. cerevisiae*, drove the fermentation. *S. paradoxus* is frequently found in association with oak trees but has been rarely isolated from fruits or fermentations [21,22], and it has only sporadically been associated with wine production [23,24]. *S. kudriavzevii* was originally isolated from decaying leaves in Japan [25] and had not been found in grape fermentations or any other type of fermentative process. However, hybrids *S. cerevisiae* x *S. kudriavzevii* were often isolated in winemaking and brewing environments [4,26,27,28]. According to González et al. [27], it is unknown whether such hybrids are the result of events that happened in the production environment or nature. Despite not naturally occurring in wine settings, *S. kudriavzevii* can conduct wine fermentation and produce the proper amounts of ethanol and glycerol by consuming the entire sugar content [28]. For instance, Arroyo-Lopez et al. [29] questioned (a) why *S. kudriavzevii* is absent from wine fermentations and (b) where their hybrids originated if *S. cerevisiae* and *S. kudiravzevii* do not coexist in wine environments. So, in a wine-model environment, it was examined how the two species competed and likely produced hybrids. They demonstrated that *S. kudriavzevii*’s lack of competitiveness, particularly at high temperatures (over 17 °C), may account for the absence of this species in wine fermentations [29]. As a result, the authors hypothesized that natural *S. cerevisiae* x *S. kudriavzevii* hybrids were more likely to have evolved in wild conditions than in industrial fermentations.

Findings from this study are the first to describe the presence of *S. kudriavzevii* in a wine environment. This species was detected, most likely because of the low fermentation temperature (9–10 °C). In fact, by comparing growth kinetics at various temperatures, it was demonstrated that only at very low temperatures (9°C) did the strains of *S. kudriavzevii*, *S. paradoxus*, and *S. cerevisiae* exhibit comparable growth kinetics; at temperatures higher than 14 °C, *S. cerevisiae* displayed better performances. These findings support those of Arroyo-Lopez et al. [29], who found that fermentation temperature plays a key role in the competitive exclusion of *S. kudriavzevii* by both *S. paradoxus* and *S. cerevisiae.*

Microfermentation experiments performed at 9°C revealed that the three strains have comparable fermentation capabilities and partially supported data previously reported by other authors [30,31] that *S. kudriavzevii* and *S. paradoxus* yield less ethanol and more glycerol compared to *S. cerevisiae*.

Wines produced with *S. cerevisiae* showed a significantly higher isoamyl acetate content and were richest in 3- + 2-methyl-1-butanol and isobutyl alcohol that, together with ethanol and even ethyl acetate, could globally act as aroma suppressors [32]. The amount of several volatile acids in *S. cerevisiae* and *S. kudriavzevii* wines were also demonstrated to vary. Higher amounts of some volatile acids were found in *S*. *kudriavzevii* wine, which may contribute to the overall aroma of the wine and, at low concentrations, have the potential to enhance the perception of fruitiness in wine due to esters or, at concentrations higher than 100 mg/L, have the opposite effect by masking fruity and varietal aromas [33,34]. Moreover, *S. kudriavzevii* wines were also the richest in beta-phenylethyl acetate and beta-phenyl ethanol, both involved in the expression of floral aromas in wines [34]. *S. paradoxus* showed an intermediate behavior between *S. kudriavzevii* and *S. cerevisiae*. This intermediate behavior showed by *S. paradoxus* is in line with data recently reported by Costantini et al. [31]. In fact, comparing the fermentation performances of *S. paradoxus* and *S. cerevisiae* on Grignolino grapes, the authors observed that the two yeast strains did not show great differences in terms of free volatile compounds in the wines at the end of the fermentation. Additionally, in partial agreement with Majdak et al. [35], *S. paradoxus* wines had a higher concentration of several volatile esters than *S. cerevisiae* samples, including ethyl butyrate, ethyl hexanoate, and beta-phenethyl acetate.

Results reveal that even though no significant differences were discovered, *S. kudriavzevii* samples always displayed consistently greater levels for the three main chemical VOC classes linked to the fermentation (i.e., total esters, alcohols, and acids).

The trend to a higher production of total esters could be explained by the fact that *S. kudriavzevii* is defined as a cryotolerant yeast and, therefore, better adapted to lower temperatures [36]. Fermentations at 9 °C could have enhanced esters production. Other studies have already investigated the ability of *S. kudriavzevii* and hybrids of *S. kudriavzevii* x *S. cerevisiae* to produce pleasant aromas in wine. It has been reported that there are many variables that can influence this process, such as nitrogen availability [37], temperature [38], fast sugar consumption during fermentation [39], and enzymatic properties [40]. In accordance with these results, other authors have observed that *S. kudriavzevii* and its hybrids produce higher amounts of fusel alcohols if compared with other yeasts. According to research by Stribny et al. [37], *S. kudriavzevii* is more efficient than other yeasts at converting phenylalanine, which may be the cause of the enhanced generation of beta-phenyl ethanol. The fusel alcohols and acetate esters content could result from the different enzymatic properties of acetyltransferase 1 and/or acetyltransferase 2 enzymes, as reported by Stribny et al. [40].

## 5. Conclusions

These findings provide information that makes *S. paradoxus* and *S. kudriavzevii* interesting yeasts for enological usage. The former is because of cryotolerance, while the latter is due to the actual trend of consumers for low-alcohol wines, as well as for its ability to produce glycerol amounts close to its taste threshold (5.2 g/L in white wine) and likely impacting sweetness and body in the mouth, low volatile acidity, and the malic acid degradation, as noted by Costantini et al. [31] and here confirmed. In this regard, additional research is required, above all in relation to varietal enology. For instance, it is reported that *S. kudriavzevii* can incorporate polyfunctional mercaptans precursor inside the cells with the potential enhancement of tropical fruit aromas [41]. Due to the significant *S. cerevisiae* competition, techniques like sequential inoculum, differential inoculum, and controlled aeration should be used when using *Saccharomyces* non-*cerevisiae* yeasts as an alternative to non-*Saccharomyces* for ethanol reduction and unique and interesting aroma development [30].

## Figures and Tables

**Figure 1 foods-12-00526-f001:**
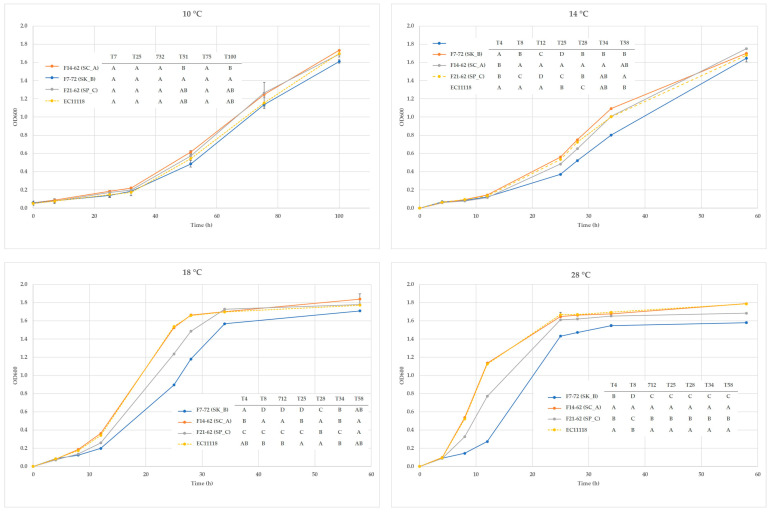
Growth kinetics at different temperatures of the three strains in YPD broth at pH 3.2. Different letters at each sampling point indicate significant differences (ANOVA: Tukey *t*-test. *p* < 0.05—XLStat).

**Figure 2 foods-12-00526-f002:**
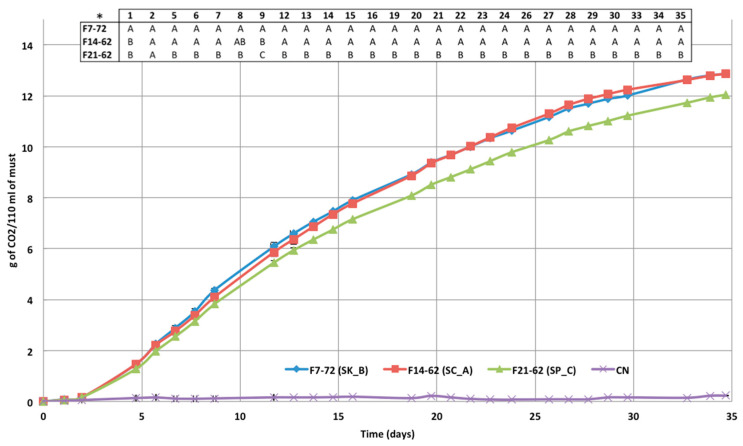
CO_2_ escape during fermentation of tyndallized Fiano must by *S. kudriavzevii* (F7-72), *S. cerevisiae* (F14-62), and *S. paradoxus* (F21-62). Data are mean values (n = 3) ± SD. * Different letters for each sampling point indicates significant differences (ANOVA: Tukey *t*-test. *p* < 0.05—XLStat).

**Figure 3 foods-12-00526-f003:**
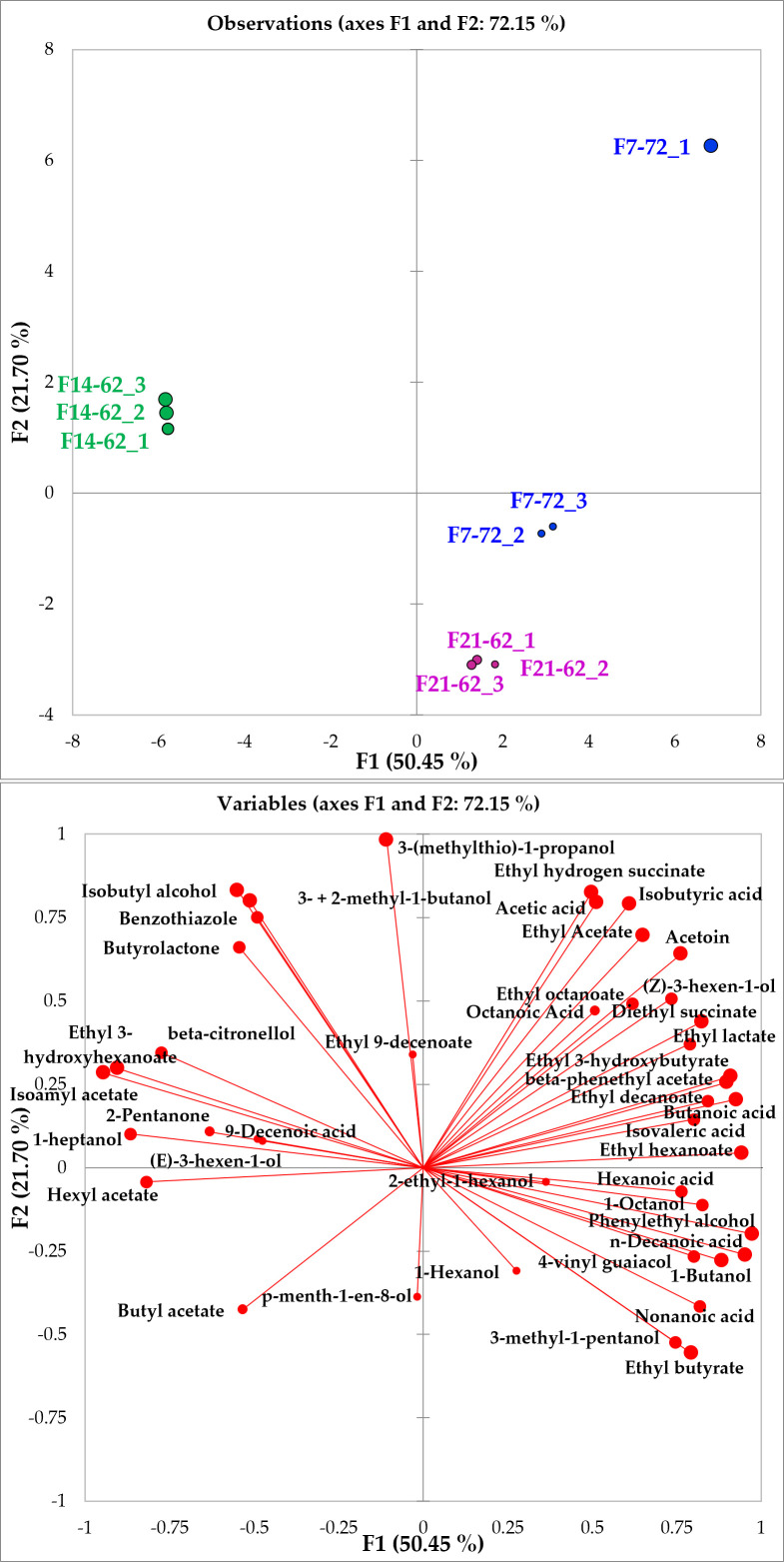
PCA biplots (Pearson correlations; α < 0.05) illustrating wine samples (**a**) and volatiles (**b**) on the first two components representing 72.15% of the total variance.

**Table 1 foods-12-00526-t001:** Yeast loads and main oenochemical parameters detected during fermentation.

Fermentation Time (dd)	pH	Total Acidity ^1^ (g/L)	°Babo	Alcoholic Degree (%*v/v*)	Yeast (Log CFU/mL)
0	3.55	9.25	22.5	0.0	5.58
7	3.69	9.07	16.2	3.2	7.77
14	3.78	8.55	5.3	10.1	7.29
21	3.91	6.26	1.5	12.8	6.88
28	3.88	6.04	0.0	13.6	6.31

^1^ Expressed as tartaric acid.

**Table 2 foods-12-00526-t002:** ITS and ITS-RFLP patterns, identification, and typing of isolated yeasts.

Isolate	ITS (bp)	ITS-RFLP Patterns			Patterns		Species	RFLP Patterns of Nuclear Genes (Restriction Enzyme)				
		*Hae*III	*Cfo*I	*Hinf*I	Interdelta	Dan4		CAT8 (*Cfo*I)	CYR1 (*Hae*III)	GSY1 (*Msp*I)	MET6 (*Hinf*I)	OPY1 (*Hae*III)
F0-51	380	280-120	200-100	^1^ nd	^2^ na	na	^3^ *M. pulcherrima*	Na	na	na	na	na
F0-52	750	750	320-310-105	nd	na	na	*H. uvarum*	Na	na	na	na	na
F7-61	380	280-120	200-100	nd	na	na	*M. pulcherrima*	Na	na	na	na	na
F7-71	850	320-230-180-150	385-365-140	365-150	A	A	*S. cerevisiae*	Nd	nd	nd	nd	nd
F7-72	850	500-230-150	385-365-140	365-150	B	^4^ nr	^1^ *S. kudriavzevii*	560-175	295-155-80-30	340-270-160	650	520-300
F7-73	750	750	320-310-105	nd	na	na	*H. uvarum*	Nd	nd	nd	nd	nd
F7-74	850	320-230-180-150	385-365-140	365-150	A	A	*S. cerevisiae*	Nd	nd	nd	nd	nd
F14-61	850	500-230-150	385-365-140	365-150	B	nr	*S. kudriavzevii*	Nd	nd	nd	nd	nd
F14-62	850	320-230-180-150	385-365-140	365-150	A	A	^1^ *S. cerevisiae*	750	560	610-160	nd	750
F14-63	850	320-230-180-150	385-365-140	365-150	A	A	*S. cerevisiae*	Nd	nd	nd	nd	nd
F14-71	850	320-230-180-150	385-365-140	365-150	A	A	*S. cerevisiae*	Nd	nd	nd	nd	nd
F21-61	850	320-230-180-150	385-365-140	365-150	A	A	*S. cerevisiae*	Nd	nd	nd	nd	nd
F21-62	850	320-230-180-150	385-365-140	365-150	C	nr	^1^ *S. paradoxus*	750	400-210	420-350	nd	520-300
F21-51	480	480	200-100-50	nd	na	na	^1^ *St. bacillaris*	Na	na	na	na	na
F21-52	850	320-230-180-150	385-365-140	365-150	A	A	*S. cerevisiae*	Nd	nd	nd	nd	nd
F28-51	850	320-230-180-150	385-365-140	365-150	A	A	*S. cerevisiae*	Nd	nd	nd	nd	nd
F28-52	850	320-230-180-150	385-365-140	365-150	C	nr	*S. paradoxus*	Nd	nd	nd	nd	nd
F28-53	850	320-230-180-150	385-365-140	365-150	A	A	*S. cerevisiae*	Nd	nd	nd	nd	nd
F28-61	850	320-230-180-150	385-365-140	365-150	A	A	*S. cerevisiae*	Nd	nd	nd	nd	nd
F28-62	850	320-230-180-150	385-365-140	365-150	A	A	*S. cerevisiae*	Nd	nd	nd	nd	nd
EC1118	nd	nd	nd	nd	nd	nd	*S. cerevisiae*	750	560	610-160	nd	750

^1^ nd: not determined. ^2^ na: not applicable. ^3^ Identification by ITS sequencing (see Table 3). ^4^ no result.

**Table 3 foods-12-00526-t003:** Results of sequencing analyses.

Strain	Gene	Species	Closely Related Accession Number	Similarity
F0-51	ITS (rDNA)	*M. pulcherrima*	NR_164379.1	100%
F0-52	ITS (rDNA)	*H. uvarum*	NR_130660.1	99%
F21-51	ITS (rDNA)	*St. bacillaris*	KY102528	100%
F14-62	ITS (rDNA)	*S. cerevisiae*	NR_111007.1	98%
F7-72	ITS (rDNA)	*S. kudriavzevii*	NR_111355.1	99%
F7-72	CAT8	*S. kudriavzevii*	LR215963.1	99%
F7-72	CYR1	*S. kudriavzevii*	LR215960.1	99%
F7-72	MET6	*S. kudriavzevii*	LR215939.1	99%
F7-72	GSY1	*S. kudriavzevii*	LR215952.1	99%
F7-72	OPY1	*S. kudriavzevii*	LR215952.1	99%
F21-62	ITS (rDNA)	*S. paradoxus*	NR_138272.1	99%
F21-62	CAT8	*S. paradoxus*	XM_033912691.1	99%
F21-62	CYR1	*S. paradoxus*	XM_033911374.1	99%
F21-62	GSY1	*S. paradoxus*	XM_033910188.1	99%
F21-62	OPY1	*S. paradoxus*	XM_033908831.1	100%

**Table 4 foods-12-00526-t004:** Oenochemical parameters of experimental wines produced with *S. kudriavzevii* (F7-72), *S. cerevisiae* (F14-62), and *S. paradoxus* (F21-62). Data are mean (n = 3) ± SD. Different letters for each parameter indicate significant differences (ANOVA: Tukey *t*-test. *p* < 0.05—XLStat).

Strain	pH	TA	VA	T-SO_2_	F-SO_2_	GF	Tac	Mac	Sac	GLY	ET
F7-72(Sk_B)	2.96 ± 0.01 _b_	6.60 ± 0.04 _b_	0.31 ± 0.02 _b_	3.18 ± 0.37 _b_	0.64 ± 0.06 _a_	1.53 ± 1.23 _b_	4.87 ± 0.06 _a_	1.10 ± 0.00 _a_	2.20 ± 0.02 _b_	5.43 ± 0.16 _a_	10.70 ± 0.89 _c_
F14-62(Sc_A)	2.99 ± 0.03 _b_	7.01 ± 0.09 _a_	0.55 ± 0.01 _a_	14.66 ± 0.041 _a_	0.82 ± 0.09 _a_	0.08 ± 0.05 _b_	5.00 ± 0.03 _a_	1.16 ±0.03 _a_	1.980.17 _b_	4.30 ± 0.09 _b_	13.53 ± 0.11 _a_
F21-62(Sp_C)	2.96 ± 0.02 _b_	6.93 ± 0.16 _a_	0.42 ± 0.01 _ab_	3.15 ± 0.05 _b_	0.74 ± 0.20 _a_	4.53 ± 0.24 _a_	4.86 ± 0.29 _a_	1.13 ± 0.06 _a_	1.22 0.78 _a_	5.54 ± 0.20 _a_	12.59 ± 0.18 _b_

TA: total acidity (g/L of tartaric acid). VA: volatile acidity (g/L of acetic acid). T-SO_2_: total SO_2_ (mg/L). F-SO_2_: Free SO_2_ (mg/L). GF: glucose plus fructose (g/L). Tac: tartaric acid (g/L). Mac: malic acid (g/L). Sac: succinic acid (g/L). Gly: glycerol (g/L). ET: ethanol (%, *v/v*).

**Table 5 foods-12-00526-t005:** Volatile compounds detected in *S. kudriavzevii* (F7-72), *S. cerevisiae* (F14-62), and *S. paradoxus* (F21-62) fermented wine samples.

RT	Compounds	F7-72 (µg/L)		F14-62 (µg/L)		F21-62 (µg/L)	
	Esters						
15.072	Ethyl butyrate	176.24 ± 20.53	a	92.79 ± 5.98	b	174.65 ± 11.50	a
17.070	Butyl acetate	2.11 ± 0.11	ns	2.82 ± 0.10	ns	2.61 ± 1.28	ns
20.093	Isoamyl acetate	226.15 ± 13.72	b	408.38 ± 8.41	a	249.70 ± 15.92	b
27.420	Ethyl hexanoate	331.46 ± 29.74	a	245.01 ± 23.79	b	318.38 ± 10.60	a
30.021	Hexyl acetate	15.47 ± 1.90	b	25.51 ± 0.51	a	21.88 ± 2.11	a
34.791	Ethyl lactate	36.66 ± 6.88	ns	30.48 ± 2.27	ns	39.34 ± 4.44	ns
40.827	Ethyl octanoate	168.64 ± 6.98	ns	132.49 ± 67.49	ns	158.36 ± 32.75	ns
46.237	Ethyl 3-hydroxybutyrate	7.37 ± 1.75	ns	3.78 ± 0.42	ns	7.14 ± 1.01	ns
53.417	Ethyl decanoate	47.42 ± 1.75	a	3.35 ± 0.21	b	33.67 ± 15.17	ab
55.624	Diethyl succinate	86.97 ± 14.32	a	30.71 ± 2.25	b	42.59 ± 3.00	b
56.451	Ethyl 9-decenoate	52.69 ± 0.11	ns	75.67 ± 64.65	ns	59.77 ± 21.83	ns
59.261	Ethyl acetate	18.69 ± 3.23	ns	13.86 ± 0.86	ns	17.65 ± 0.44	ns
63.347	beta-Phenethyl acetate	114.46 ± 4.54	a	19.56 ± 0.83	c	53.02 ± 2.33	b
67.500	Ethyl 3-hydroxyhexanoate	2.72 ± 0.65	b	6.88 ± 0.09	a	3.20 ± 1.15	b
90.526	Ethyl hydrogen succinate	74.97 ± 6.18	ns	70.37 ± 30.17	ns	50.17 ± 4.85	ns
	Tot Esters	1362.02 ± 55.67	ns	1161.65 ± 108.93	ns	1232.13 ± 22.96	ns
	Alcohols						
18.187	Isobutyl alcohol	303.62 ± 20.18	b	705.87 ± 26.73	a	192.20 ± 26.81	b
21.581	1-Butanol	31.18 ± 2.17	b	4.83 ± 0.41	c	67.99 ± 9.71	a
25.962	3- + 2-methyl-1-butanol	10,771.47 ± 307.35	ab	13,128.03 ± 454.11	a	10,162.02 ± 817.31	b
33.910	3-methyl-1-pentanol	13.48 ± 1.88	b	6.08 ± 0.77	c	20.62 ± 1.26	a
35.670	1-Hexanol	733.39 ± 11.72	ns	737.89 ± 14.98	ns	792.96 ± 63.41	ns
36.315	(E)-3-hexen-1-ol	4.48 ± 0.85	ns	5.83 ± 1.13	ns	5.40 ± 0.41	ns
37.671	(Z)-3-hexen-1-ol	22.99 ± 0.28	ns	20.02 ± 0.40	ns	20.26 ± 1.29	ns
42.352	1-heptanol	26.81 ± 1.58	b	38.17 ± 0.93	a	32.49 ± 1.96	ab
44.510	2-ethyl-1-hexanol	5.58 ± 1.14	ns	5.06 ± 0.65	ns	5.70 ± 1.40	ns
48.764	1-Octanol	6.59 ± 0.51	a	1.28 ± 0.78	c	3.56 ± 0.60	b
68.579	Phenylethyl Alcohol	7802.38 ± 224.25	a	4023.73 ± 44.05	b	7102.09 ± 372.12	a
	Tot Alcohols	19,721.96 ± 115.92	ns	18,676.77 ± 527.82	ns	18,405.28 ± 1173.81	ns
	Acids						
41.747	Acetic acid	271.48 ± 184.28	ns	303.22 ± 27.87	ns	227.73 ± 32.15	ns
49.165	Isobutyric acid	18.60 ± 0.93	a	8.96 ± 1.47	b	5.68 ± 0.62	b
52.765	Butanoic acid	2.81 ± 0.46	ab	0.78 ± 0.40	b	3.91 ± 1.54	a
55.284	Isovaleric acid	43.81 ± 7.99	ns	31.42 ± 9.61	ns	52.93 ± 9.05	ns
64.966	Hexanoic acid	353.46 ± 18.34	ns	241.87 ± 66.07	ns	328.66 ± 59.87	ns
75.863	Octanoic Acid	1169.95 ± 63.35	ns	807.08 ± 127.31	ns	717.04 ± 127.28	ns
80.932	Nonanoic acid	5.16 ± 0.23	a	0.00 ± 0.00	b	3.81 ± 1.21	a
85.752	n-Decanoic acid	452.59 ± 47.03	a	15.66 ± 8.13	b	388.58 ± 25.67	a
88.443	9-Decenoic acid	443.78 ± 55.42	ns	588.22 ± 91.65	ns	547.06 ± 78.08	ns
	Tot Acids	2761.64 ± 323.98	ns	1997.21 ± 299.11	ns	2275.40 ± 181.07	ns
	Miscellaneous						
12.166	2-Pentanone	7.73 ± 1.61	a	8.03 ± 0.44	a	3.45 ± 0.43	b
30.739	Acetoin	87.82 ± 10.32	a	25.28 ± 0.73	b	39.21 ± 2.33	b
52.572	Butyrolactone	30.49 ± 4.63	b	41.28 ± 0.57	a	33.16 ± 2.07	b
56.949	p-menth-1-en-8-ol	4.99 ± 1.07	ns	5.25 ± 0.73	ns	5.80 ± 0.36	ns
58.048	3-(methylthio)-1-propanol	25.95 ± 0.22	b	37.08 ± 1.51	a	21.16 ± 1.68	b
60.817	beta-citronellol	5.96 ± 0.81	ab	7.21 ± 0.45	a	5.42 ± 0.45	b
70.661	Benzothiazole	7.93 ± 0.41	ab	14.76 ± 4.72	a	0.00 ± 0.00	b
82.159	4-vinyl guaiacol	25.87 ± 7.14	a	2.38 ± 0.39	b	39.98 ± 1.29	a
	Tot Miscellaneous	196.74 ± 6.66	a	141.26 ± 5.04	b	148.19 ± 4.86	b

Different letters in the row of each compound refer to significant differences (Tukey, *p* ≤ 0.05); ns = not significant.

## Data Availability

Data is contained within the article.

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
