# Peer review of "Isolation and Characterization of Cryotolerant Yeasts from Fiano di Avellino Grapes Fermented at Low Temperatures"

_foods, 2023, doi:10.3390/foods12030526_

Round 1

Reviewer 1 Report

Dear Authors, In this study, S. kudriavzevii and S. paradoxus as novel cryotolerant yeast strains were isolated and investigated their fermentative performances and volatiles production. This paper reports a very interesting approach. I had some doubts that I would like you to clarify: Introduction section: This section should set the context for the study, clearly state the research objective, and establish the significance of the study. GS/MS: The concentrations of volatile compounds are by reference to a single internal standard (2-octanol). It has been assumed that the GC (FID) responses for the variety of volatile components present are the same as that for the internal standard. This is not so, as GC (FID) responses vary widely depending on the compounds. What is the recovery yield of the internal standard? It is not clear how the quantitative analysis was conducted. Were the reference standards of all compounds used in the identification/quantification? Please state the source of the reference standards. No data are reported about NaCl use. Because the salting-out effect should move volatile solutes to the headspace when the headspace sampling technique was applied. Is it used or not used? Sensory profile: Have you performed a sensory test? A correlation with GC/MS can be stated. Figures: The figures and graphs should present data clearly.

Author Response

Dear Authors, in this study, S. kudriavzevii and S. paradoxus as novel cryotolerant yeast strains were isolated and investigated their fermentative performances and volatiles production. This paper reports a very interesting approach. I had some doubts that I would like you to clarify:

We appreciate that you considered our work to be worthwhile.

Introduction section This section should set the context for the study, clearly state the research objective, and establish the significance of the study.

Research objectives and significance of the study were enhanced by adding missing details at lines 49-62.

GS/MS: The concentrations of volatile compounds are by reference to a single internal standard (2-octanol). It has been assumed that the GC (FID) responses for the variety of volatile components present are the same as that for the internal standard. This is not so, as GC (FID) responses vary widely depending on the compounds. What is the recovery yield of the internal standard? It is not clear how the quantitative analysis was conducted. Were the reference standards of all compounds used in the identification/quantification? Please state the source of the reference standards.

We appreciate the referee's comment. We are conscious that the method we used is a semiquantitative method, so results are only presented and discussed by comparing experimental samples to one another in order to find significant differences using ANOVA, never by guessing at VOCs concentrations

The use of a semiquantitative approach is entirely appropriate for this type of approach, and it is widely used in the scientific literature (Giordano et al., 2009 - doi: 10.20870/oeno-one.2009.43.3.796; Sáenz-Navajas et al., 2010, doi: 10.1021/jf904377p; Rodrguez-Bencomo et al., 2011, doi: 10.1002/jsfa.4494; Caven-Quantrill et al. 2017 – doi:10.3390/beverages3040062;  De Filippis et al. 2019 - doi: 10.1016/j.foodres.2018.11.033; Moreno-Olivares et al. 2020 – doi:10.1002/jsfa.9991; Bianchi et al. 2023 – doi: 10.1016/j.foodchem.2022.134138). Nevertheless, in order to address this comment, we specifically mentioned this element in the "Materials and procedures" section (paragraph: 2.6. VOCs analysis – lines 146-147 and 157-160).

No data are reported about NaCl use. Because the salting-out effect should move volatile solutes to the headspace when the headspace sampling technique was applied. Is it used or not used?

NaCl was not used since a liquid-liquid extraction method was used instead of a headspace isolation strategy (e.g., SPME). To avoid this misunderstanding, we stated explicitly that liquid-liquid extraction was used to isolate VOCs in the "Materials and procedures" section (paragraph: 2.6. VOCs analysis - lines 146-147).

Sensory profile: Have you performed a sensory test? A correlation with GC/MS can be stated.

The wines under investigation did not undergo sensory analysis, nor was this type of analysis ever reported to have occurred.

Figures: The figures and graphs should present data clearly.

The readability of the figures 1 and 3 was improved.

Reviewer 2 Report

This paper identified and characterized natural cryotolerant yeast strains isolated from Fiano white grapes, and compared their fermentation capabilities and flavor substance formation characteristics. The findings in this study provided some new insights for the Fiano wines fermentation, and the obtained strains are interesting yeasts as an alternative to non-Saccharomyces for ethanol reduction and for unique and interesting aroma development. The manuscript is well organized and results are convincing, however, in my opinion, the current version manuscript required to be modified according to the following suggestions.

1. The title of the article needs to be reconsidered. In fact, the yeast strains studied in this manuscript were isolated from cultures at different stages of winemaking, not the fresh grape raw materials.

2. The background introduction seems too shorter or concise, it is better to provide more research details of references, and state the main experiment content and significance and value of this study in the last paragraph of the introduction section.  .

3. The culture formula of the LW-nutrient agar (Line 73) should be addressed.

4. For the abbreviation “RFLP” in line 90, should be defined at first mention. 

5. The gene sequencing results need to be submitted to the relevant international database such as NCBI and obtain their registration number, and added to Table 3.

6. The text font size in Figure 1 is too small, and not clear enough for the readers, please modified them carefully.

7. In the Table 4 and table 5, letters representing significant difference results should be changed to lower case style.

Author Response

This paper identified and characterized natural cryotolerant yeast strains isolated from Fiano white grapes, and compared their fermentation capabilities and flavor substance formation characteristics. The findings in this study provided some new insights for the Fiano wines fermentation, and the obtained strains are interesting yeasts as an alternative to non-Saccharomyces for ethanol reduction and for unique and interesting aroma development. The manuscript is well organized and results are convincing, however, in my opinion, the current version manuscript required to be modified according to the following suggestions.

We appreciate that you considered our work to be worthwhile.

  1. The title of the article needs to be reconsidered. In fact, the yeast strains studied in this manuscript were isolated from cultures at different stages of winemaking, not the fresh grape raw materials.

Since grapes were harvested, processed, and fermented under sterile circumstances, isolated strains had to come from grapes. In any case, the title was modified to prevent confusion.

  1. The background introduction seems too shorter or concise, it is better to provide more research details of references and state the main experiment content and significance and value of this study in the last paragraph of the introduction section.

Research objectives and significance of the study were enhanced by adding missing details at lines 49-62.

  1. The culture formula of the LW-nutrient agar (Line 73) should be addressed.

Composition was added at lines 83-87.

4.For the abbreviation “RFLP” in line 90, should be defined at first mention.

The definition was added. Please see line 99.

  1. The gene sequencing results need to be submitted to the relevant international database such as NCBI and obtain their registration number, and added to Table 3.

Table 3 was improved by the addition of the sequences’ closest accession numbers. No sequence was required to be sent because the similarity was always greater than 98%, but we can if the reviewer demands it.

  1. The text font size in Figure 1 is too small, and not clear enough for the readers, please modified them carefully.

Figure 1 has undergone major revisions, which should improve readability.

  1. In the Table 4 and table 5, letters representing significant difference results should be changed to lower case style.

Both tables' uppercase characters were changed, and table 5 was also improved.

Round 2

Reviewer 1 Report

Considering the reviewers' suggestions, the authors improved the quality of this work.

Reviewer 2 Report

The repsent manuscript has been improved.